# A Hybrid Data-Based and Model-Based Approach to Process Monitoring and Control in Sheet Metal Forming

**Sravan Tatipala \***[ID]**, Johan Wall**[ID]**, Christian Johansson**[ID]** and Tobias Larsson**[ID]

Department of Mechanical Engineering, Blekinge Institute of Technology, Valhallavägen,
371 41 Karlskrona, Sweden; johan.wall@bth.se (J.W.); christian.johansson.askling@bth.se (C.J.);
tobias.larsson@bth.se (T.L.)
\* Correspondence: sravan.tatipala@bth.se

**Abstract:** The ability to predict and control the outcome of the sheet metal forming process demands holistic knowledge of the product/process parameter influences and their contribution in shaping the output product quality. Recent improvements in the ability to harvest in-line production data and the increased capability to understand complex process behaviour through computer simulations open up the possibility for new approaches to monitor and control production process performance and output product quality. This research presents an overview of the common process monitoring and control approaches while highlighting their limitations in handling the dynamics of the sheet metal forming process. The current paper envisions the need for a collaborative monitoring and control system for enhancing production process performance. Such a system must incorporate comprehensive knowledge regarding process behaviour and parameter influences in addition to the current-system-state derived using in-line production data to function effectively. Accordingly, a framework for monitoring and control within automotive sheet metal forming is proposed. The framework addresses the current limitations through the use of real-time production data and reduced process models. Lastly, the significance of the presented framework in transitioning to the digital manufacturing paradigm is reflected upon.

**Keywords:** in-line measurement data; modelling and simulation; product quality; process performance; process monitoring and control; Industry 4.0; sheet metal forming

---

## 1. Introduction

The sheet metal forming (SMF) process involves non-stationary conditions and complicated phenomena such as non-linearities, temperature variation, batch-to-batch fluctuations in material properties, and complex product geometries, which makes it challenging to achieve desired product specifications and ensure process performance [1–5]. Due to the high tooling costs associated with SMF, justified by large-volume and efficient production runs, product quality control is of high importance [6]. Timely detection, diagnosis (identification of root-cause), and correction of a fault are paramount to enhance process performance, which leads to improved product quality and reduced production costs.

In the past, most industrial process monitoring (IPM) approaches were focused on fault detection, i.e., on the ability to detect a fault and reduce the time between a faults' occurrence and detection [7]. More recently, with concepts like zero-defect manufacturing gaining importance, the focus has shifted toward fault diagnosis and troubleshooting activities that consume a considerably larger portion of the process downtime [1,7] compared to fault detection activities. In this context, several data-driven [7–11], model-based [12–14], and statistical [15] approaches have been proposed to support

the identification of the underlying root cause of a fault. However, most of these approaches lack features necessary to completely diagnose and isolate a fault [16]. Similarly, numerous control approaches have been developed to regulate increasingly complex manufacturing processes [17,18]. Although these approaches have gained significant popularity within sophisticated manufacturing processes such as in semiconductor production or chemical plants, their potential within other manufacturing domains such as metal forming remains underexploited [18]. Most control approaches within metal forming aim to control and optimize machine parameter settings instead of the product properties, which is usually the property of interest for the end customer [6,19]. Furthermore, the control actions in such approaches are generally based on either offline models or online process models, which involve several approximations to be computationally feasible [19]. In both cases, the influence of product-process parameter correlations on the output product quality is overlooked by not explicitly modelling such complex relationships [20]. Thus, in the presence of changing production conditions, the overall effectiveness of control loops is affected, which leads to higher costs and process downtime. To summarize, the prevalent monitoring approaches lack features to comprehensively analyse and isolate parameter(s) causing process upsets [7]. Similarly, control approaches within complex automotive processes such as sheet metal forming, although effectively controlling production tool settings [19], encounter challenges due to varying production conditions, which leads to inconsistent output product quality and process faults.

One way to deal with this issue is by combining data-driven and model-based approaches into a hybrid procedure [7,19]. Data-driven refers to approaches that extract the necessary information for the underlying process model structure predominantly from operation data (in addition to subsidiary sources such as background process knowledge, process flowsheets, etc.). Industry 4.0 [21] promises new opportunities in this context, which would facilitate collection, transmission, and exploitation of production data to adapt/improve the manufacturing processes. Model-based refers to approaches where the underlying model structure is based on prior knowledge of the process and its behaviour, which is deduced from first-principle models. These incorporate a very detailed process-specific structure.

The current work proposes a framework for enhancing monitoring and control of sheet metal forming through a hybrid data-based and model-based approach. The proposed framework uses data from production to capture the observations/deviations present in the real process while the process model helps to interpret this data using knowledge of the process behaviour derived from advanced physics-based simulation models. The guiding research question of the paper is:

RQ: What are the shortcomings of the process monitoring and control approaches within sheet metal forming and how can these be mitigated?

## 2. A Theoretical Basis for Process Monitoring and Control

Monitoring and control of manufacturing processes are challenging due to the wide number of process parameters involved in shaping the output product characteristics [19]. Broadly speaking, the topic of monitoring and control of complex manufacturing processes can encompass methodologies at several levels within the manufacturing realm, e.g., production process and equipment level, manufacturing systems-level, and production operation planning level. The current paper limits its scope for the production process and equipment level. The following section provides an overview of the methodologies employed at the production process and equipment level.

### 2.1. Process Monitoring

Industrial Process Monitoring (IPM) is an activity that involves tracking of machinery, systems, and processes in real-time to achieve production consistency, safety, and economic viability [7]. The focus of IPM has evolved through the years, initially almost entirely dedicated to improving fault detection performance while, more recently, diagnosis and root-cause analysis is gaining importance in which the set of variables causing a fault are identified and isolated [16]. Moving forward, the

emphasis is in the direction of predictive analytics where the evolution of operational risks is assessed, which aids better planning and shutdown operations, minimizes production losses, and secures equipment/operator safety [7].

Reis [16] presents a review on the prevalent process monitoring approaches highlighting recent trends within IPM and the variety of approaches emerging within applied statistics, engineering, and machine learning communities. The review points out that classical monitoring approaches, which incorporate a generic probabilistic model structure representing normal operating conditions (NOC), such as Shewhart [22], Exponentially Weighted Moving Average (EWMA), and Cumulative Sum (CUMSUM) [23], are unsuitable for root cause analysis as these methods do not contain process-specific structure information other than the parameter estimates of the NOC model obtained from process data [7]. Modern industrial settings are characterized by increasing data volumes and dimensions, non-stationary dynamics, heterogeneous datasets, and monitoring of the correlation structure. As a consequence, monitoring approaches have evolved to multivariate approaches and further to high-dimensional methodologies such as Principal Component Analysis (PCA) and Partial Least Square (PLS) [7]. However, these approaches are non-causal and, hence, would lead to ambiguous results when used for fault diagnosis purposes [16]. A good example is the smearing-out effect in PCA-based process monitoring [24–26] in which, for a given change in the monitoring statistic, the method is not able to distinguish between the set of variables causing a fault and the variables that are a consequence of the fault.

As a response to this limitation, subsequent approaches aimed at incorporating more of the causal information into process NOC models, i.e., the knowledge regarding causes and effects of a change in state are embedded into the underlying model structure to effectively guide fault diagnosis and troubleshooting activities. A common perception among process monitoring research [7,16,27] is that the inclusion of process-specific information into the monitoring methods shall bring extra sensitivity toward process fluctuations leading to finer fault detection/diagnosis. Though this will benefit the fault detection capability, fault diagnosis/troubleshooting activities will reap significantly greater benefits from the added sensitivity since these activities are known to consume a considerably larger portion of the process downtime [7]. The approaches in this context may be divided into two categories, which are data-driven and model-based categories. Some examples of the data-driven and model-based approaches in this context can be found in References [7,16].

### 2.2. Process Control

Process control, in simple terms, is the activity of regulating a process using an automated control system based on process measurements. Several publications have reviewed the state of the art in the advancements of a manufacturing process control at a generic level spanning multiple process control techniques [17,18,28] or within specific application areas [6,29]. Part quality and dimensional integrity are two important factors governing the productivity of machine tools especially in high-speed regimes [17,28] and, hence, several strategies for control of servo-systems, estimating contour error, tool path profile planning, feed-rate scheduling for cycle time optimization, tool condition monitoring, and overall productivity optimization have been developed [17].

In contrast to the automatic feedback control on sensed signals, the approach of adjusting process inputs to the predicted responses of the process is broadly referred to as model-based process control [17]. Popular control strategies in this regard include adaptive process control, model predictive control, time-delay control, long-term process control, etc., as elaborately explained in Reference [17]. The interested reader is referred to References [17,18] for more information. Most of these methods concentrate on the control of a single operation as an independent manufacturing process [17]. However, a wide range of industries has multi-stage manufacturing processes (e.g., automotive, semiconductor), which involves interactions between sequentially executed process operations and across the operation on a particular product [18]. Furthermore, the need to understand and control such interactions is recognized by Reference [17]. Likewise, statistical process control [30] has been

popular within industries including multi-stage manufacturing processes and has led to huge gains [31]. However, it does not prescribe corrective measures when quality drifts are observed [31]. This led to the inception of run-to-run (R2R) control almost three decades ago [18]. The application of R2R control encompasses several industries while a major portion of academic research in this field is conducted within the semiconductor manufacturing industry [17]. The approaches mentioned thus far, although generally popular within industries like semiconductors, pharmaceutical, machining, and some types of automotive manufacturing (e.g., assembly process) have limited use within the metal forming processes.

Table 1 summarizes the identified challenges of data-driven and model-driven approaches, respectively, in the context of process monitoring and control while also highlighting merits and demerits.

**Table 1.** Summary of identified challenges of prevalent methods in the context of process monitoring and control.

| | Data-Driven Approaches | Model-Driven Approaches |
|---|---|---|
| **Issues/Challenges** | • Large quantities of data required for reliable performance<br>• Collected data is limited to confined operating conditions<br>• Data imbalance, i.e., varying amounts of data from different components in the process | • Rigorous efforts needed for process model derivation<br>• Accuracy versus computational cost<br>• Model fidelity<br>• Do not cater to disturbances in reality |
| **Merits** | • Require minimal knowledge of process behaviour<br>• Cater to the disturbances in reality<br>• Favourable in big data situations | • Include cause-effect relationship knowledge |
| **Demerits** | • Lacks causal structure<br>• Dimensionality issues<br>• Susceptible to smearing-out effect<br>• Lacks features to fully diagnose/isolate fault cause | • Lacks ability to handle noise, which is prevalent in real production scenarios |

## 3. Process Control within Metal Forming: Challenges and Opportunities

Almost three decades ago, Reference [32] reviewed closed-loop control of product properties recognizing the restricted influence of the control community has had on the progress of manufacturing processes or process control despite tremendous potential. Around a decade ago, Reference [29] presented an exhaustive review on the advances in the control of SMF. The review classifies control procedures to tackle issues within SMF into four categories: open-loop control, closed-loop machine control, in-process control, and cycle-to-cycle control. Open-loop control is a pre-process procedure to adjust tooling design (through grinding and welding) and the process variables in the tool try-out section before using it in the production line. This procedure is time-consuming and involves several trials and a poorly tuned SMF process leads to frequent production process halts. Closed-loop machine control focuses upon in-line control of SMF tool forces either using process models or die-maker experiences. The main disadvantage of this is that it cannot handle disturbances (e.g., variation in lubrication and blank thickness) affecting part quality and process consistency [33]. Cycle-to-cycle control uses Statistical Process Control (SPC) methods to implement control. Such control maintains a

database of important process variables. This database is used by machine operators to determine critical process variables and manually adjust the process. In-process control ensures that a process variable follows a reference trajectory by manipulating tool forces. Although in-process control is a reasonable solution for overcoming production inconsistencies, there are still many challenges with regard to the design of the process controller, while determining reference trajectory as well as sensing/actuation technology development [29].

Most metal forming processes today have control systems and multiple feedback loops [19]. The majority of these control systems aim to control the equipment and not the properties of the actual product [19], which is usually the main property of interest for customers. For example, a control loop may regulate the force or speed of a machine tool, which, in turn, contributes to shaping the output product. However, it is uncommon to have unique feedback loops for regulating, for instance, the temperature or the residual stresses in the workpiece that directly influences the microstructure of the output product [19]. As a consequence, the influence of disturbances due to variation in material properties, the lubrication amounts and the blank thickness cannot be eliminated [6,29]. Two tests involving closed-loop control was conducted by Reference [33] with a predetermined blank holder force trajectory while different lubrication conditions show differences in part quality.

Likewise, Julian M. Allwood et.al. [6] argue that current control strategies, although claiming to be closed-loop control, focus to match the tool settings to a pre-planned target sequence rather than iteratively updating the tool settings based on the current state of the workpiece so as to match the output product state to the desired product state. Moreover, these control systems are based on off-line process models where the control systems ensure the operation of the equipment according to a model developed before the operation begins [6]. According to Reference [6], three key aspects to convert the current control approaches within metal forming to be proactive are: sensors (types of sensors for monitoring workpiece during process operation), actuators (introduction of actuators with sufficient flexibility and responsiveness to allow adjustments in response to evolving process conditions), and models (creation of models of sufficient accuracy and speed to allow operation online).

## 4. Research Approach

The research presented in this paper is part of a research project that is structured according to the Design Research Methodology (DRM) [34]. DRM is chosen because it allows the researchers to take the scientific investigations of the research challenges into a solution-oriented mode, where the point is to devise solution proposals that aim to remedy the problem being investigated. In detail, DRM is structured as an iterative four-stage process that has similarities with a product development process but emphasizes the scientific approach. The process begins with a clarification of the research objectives before researchers are expected to investigate and define the research problem. With this in perspective, the researchers then propose solutions that should improve the identified problem. The final step of the research consists of conducting comprehensive studies to evaluate the efficacy of the proposed support. With this structure in place, researchers employ different data collection methods that are common with other research methodologies, which are appropriate for the investigation.

Starting from a research objective of modelling and predicting product properties in-line with the SMF-process and, thus, tune the process parameters while it runs piece-by-piece to obtain desired outputs, this paper builds upon previous qualitative research [5,35,36] that was conducted with a global automotive manufacturer where challenges with integrating measured data with in-line modelling and simulation was investigated. Based on the findings from these empirical studies, this paper aims to develop a framework for how to integrate and automate the different solution components into a hybrid control and modelling approach.

Papers for literature review were collected through *Scopus* and *Google Scholar* databases, including proceedings of conferences related to the research context. The snowballing technique [37] was used for finding additional relevant literature. Initially, state of the art (SOTA) papers in the domain of monitoring and control of manufacturing processes were studied to gain an overarching summary of the

methods employed for dealing with product quality issues and process upsets while understanding the challenges involved. Furthermore, approaches within the application domain of SMF were examined to identify gaps and opportunities. The review included papers written in English and published in relevant journals. The papers were filtered by reading the title, abstract, and conclusion in the initial stage to judge its relevance to the research context followed by reading the full-text. The list of selected papers was thoroughly read, analysed, and the main points were extracted. Lastly, based on the previous research [5,35,36], together with the literature review conducted in the current paper, a framework for monitoring and control within automotive SMF is proposed.

## 5. Results

### 5.1. Motivation for the Framework

To tackle the shortcomings of the data-driven or model-based approaches, several researchers have made recommendations on the possible way forward. Reference [6] points toward the need to consider the current state of the system while determining control response instead of utilizing a pre-planned control sequence built using offline process models. Furthermore, Reference [20] foresees the development of metal forming research toward a hybrid approach where physics-based models are combined with big data in manufacturing. On similar lines, Reference [19] points to the potential of providing significant improvements in the quality of a finished product through the combination of increased information about product properties and model-based control systems. Additionally, Reference [29] envision the creation of systems that will combine Statistical Process Control (SPC), machine control, in-process control, and cycle-to-cycle control capabilities to significantly improve the part quality and consistency within the SMF process. In short, research [6,7,19,20,29] predicted the potential of benefiting from a combination of data-and-model-based approaches moving forward. In general, there is a lack of an overarching framework for monitoring and control of complex manufacturing processes, with only a few examples depicting application-specific approaches such as shown in References [6,38]. One explanation for this, as pointed out by Reference [19], is the presence of a vast range of manufacturing processes and products, which makes it difficult to propose such an approach.

### 5.2. Instantiating the Framework

The proposed framework, combining data-driven and model-based concepts into a hybrid methodology, is presented in Figure 1. The framework consists of the following steps. It starts with an input to the manufacturing line, which is uniquely marked. Through that marking, the input is identified, and input specification data is retrieved from data storage (e.g., material properties and surface coating), while other input data (e.g., temperature, thickness, and friction conditions), which are not available from data storage, are measured using sensors in the process. The retrieved and measured data, together with the desired target product specifications, are fed into process models to proactively predict the product/process settings needed to attain the desired output.

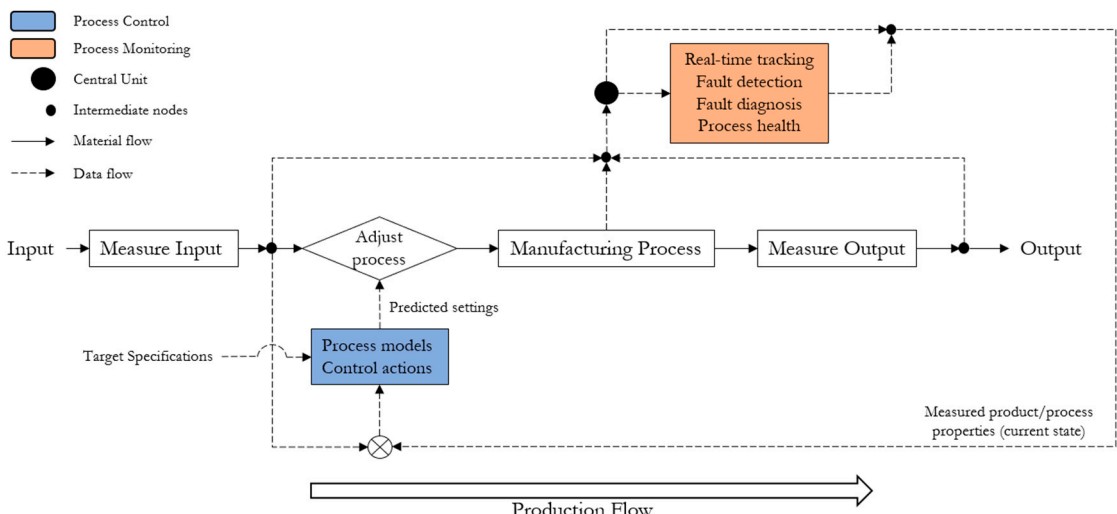

**Figure 1.** Schematic of the proposed framework.

Furthermore, the predicted settings are used to adjust the product properties (e.g., friction conditions and temperature) and/or process tool settings (e.g., force, speed, and displacement). Using these settings, the manufacturing process executes its operation. Depending on the type and complexity of the part being produced, there may be several manufacturing steps. This would then imply that the product/process settings need to be adjusted at several intermediate steps in a similar manner. After the manufacturing process finishes execution, sensors measure the output product quality (e.g., surface finish, geometry, and dimensions). All process sensors continuously stream data to a central unit (in addition to intermediate nodes).

This data is used by the monitoring platform to supervise product/process variables, detect faults, identify the root-cause of faults, and to maintain process health. The monitoring platform includes data-driven models (primarily for variable supervision, fault detection, and process health monitoring), hybrid models (a combination of data-driven and model-based models for diagnosis and predictive analytics) and contains information about normal operation conditions (NOC). Depending on the scenario, the extent of data possible to harvest, data storage/transmission infrastructure, and type of process models as well as the effort required to derive them, which varies data-driven/model-based methods, are suitable.

Likewise, the information from the monitoring platform and the measured data is fed into the process control system. The control system uses reduced process models (e.g., meta-models [13,14], for viability in real-time) to determine an appropriate control response for diminishing the gap between the actual process output and the desired process output. These settings are used to adjust the manufacturing process. The framework includes a feed-forward control loop, a feed-back control loop, and a monitoring platform. The feed-forward control loop adjusts the process based on measured data as well as data retrieved from data storage. The feed-back control loop adjusts the process based on the current system status, i.e., in-line measured product properties and/or process tool settings along with measured output product quality (e.g., surface finish, geometry, and dimensions). The monitoring platform enhances product/process supervision and process fault detection/diagnosis while providing product/process current state information to the control system.

## 6. Discussion

On one hand, data-driven approaches benefit by requiring a minimal understanding of the inherent process mechanisms [8] while, on the other hand, suffer from dimensionality issues, which often requires a huge amount of data for reliable performance and do not contain features for full fault diagnosis and isolability [7]. On the other hand, model-based approaches, albeit capture, provide a clear cause-effect relationship between product/process parameters [11]. They fail to take into account

the external disturbances and noises prevalent in real production processes [8] while also require rigorous experimentation/effort to derive the process models [6]. It is believed that the combination of operation data that capture fluctuations present in reality with model-based approaches that comprise of extensive knowledge regarding product/process behaviour could benefit the monitoring and control approaches in the context of complex manufacturing processes. Current research, while reviewing research efforts in this context, instantiates a hybrid framework for enhancing process monitoring and control within automotive SMF.

Moving forward, several practical considerations are of interest and important to highlight. Several industrial sectors (e.g., chemical, pharmaceutical, semiconductors, food [7]) have already started reaping the benefits from data. These opportunities are still quite under-exploited in automotive SMF, especially due to the complexity, scale, high tooling costs, and advanced process control involved. Even though literature clarifies the presence of several promising concepts related to sensor, tooling, and modelling [6] within metal forming, it has led to little industrial take-up [6,20], which depicts the difference between state-of-the-art and state-of-practice. Furthermore, solely collecting data in itself does not generate benefits. Usually, a large data imbalance exists in data accumulated from different manufacturing scenarios (e.g., normal operating conditions and faulty examples), which creates challenges with regard to accurate sample density estimations and construction of appropriate class boundaries for multi-class classifier approaches [12]. Similarly, the incompleteness of data (e.g., missing or erroneous values) often common in large-scale industrial processes could lead to inaccurate computations (projections and predictions) and, hence, research related to incomplete data is an interesting area [27]. Likewise, most large-scale industrial processes involve challenging industrial conditions (e.g., noise, dirt, and vibrations) and, therefore, need more research to improve the effectiveness of the approaches in such environments. Issues also stem from dimensionality, heterogeneity, and the presence of multiple data management tools within such processes. Thus, although increasing data volumes create new possibilities to drive production processes to higher performance levels, there is a lack and need for newer strategies to enable that possibility [39].

To realize the proposed framework, it is paramount to establish effective communication between the various components in the manufacturing environment. Internet of Things (IoT) technology would provide access to a central repository [6] for managing data storage, access, and transmission to the required regions. Likewise, data processing and automation strategies for analysing, interpreting, and configuring operation data are equally essential to realistically manage huge amounts of data, render useful insights, and foster connectedness among various parts of the production plant [5,35,36].

Despite today's computing power, finite element models for sheet metal forming applications remain computationally expensive and, as such, not suitable for use in online closed-loop control [6]. Hence, most model-based control efforts have resorted to approximations to reduce the solution time even though the developers of process models have been focusing on accuracy rather than increasing speed [6]. Furthermore, deriving appropriate process model reductions based on an accurate simulation model involves a lot of modelling and validation efforts to produce good results [14]. Thus, more efforts toward effectively deriving reduced process models are needed.

To some extent, the proposed framework resembles that of a digital twin [40]. In the authors' opinion, an implementation of the proposed framework may form the basis of a digital twin considering the connectivity, sensor setup, process model, and data analysis included. However, more analysis tools and a stronger focus on predictive capabilities should be added for it to be a fully functional digital twin. Furthermore, it should also be emphasized that moving toward a digitized process control is a strategic rather than a tactic decision. It needs to be evaluated and assessed concerning overarching company strategies regarding the product portfolio as well as production capabilities and needs.

## 7. Conclusions

Most modern metal forming operations include control systems and multiple feedback loops. However, these are predominantly focused on controlling the machine settings while aiming to

move toward controlling product properties. In this context, several data-driven and model-based approaches are prevalent, and it is believed that a combination of these approaches helps eliminate the shortcomings of the current approaches. The current paper introduces a framework for monitoring and control within automotive SMF implemented through a hybrid data-based and model-based approach. A hybrid approach benefits by capturing the disturbances prevalent within the real production process while also incorporating extensive knowledge about system behaviour and parameter correlations.

To facilitate the implementation of the framework, the research recognizes the need to bring monitoring and control realms closer together. Such hybrid approaches, in the authors' opinion, will be paramount in establishing connectedness within an industry 4.0 enabled manufacturing process. Concepts facilitating data collection, transmission, and exploitation to adapt/improve the manufacturing processes seem promising regarding the framework applicability. The proposed framework is believed to also be relevant to complex manufacturing processes in general.

Future work will focus on research toward assessing the validity of the approach and judging its feasibility via the implementation of the framework on a lab-scale prototype environment. This would help in further developing the framework providing deeper insights into other practical problems to be considered during large-scale implementation. In the future, it would also be interesting to explore how the knowledge obtained from production processes can be useful for designing better products in the early stages of product development as a means of getting design and manufacturing closer.

**Author Contributions:** Conceptualization, S.T., J.W., C.J.; Methodology, S.T.; Formal Analysis, S.T.; Investigation, S.T.; Data Curation, S.T.; Writing-Original Draft Preparation, S.T.; Structuring, Rewriting, Recurrent feedback and Discussions, J.W., C.J., T.L. All authors have read and agreed to the published version of the manuscript.

**Funding:** The research leading to these results has received financial support from the Swedish Knowledge and Competence Development Foundation (Stiftelsen för Kunskaps-och kompetensutveckling) through the Model Driven Development and Decision Support (MD3S) research profile at Blekinge Institute of Technology and via Test-arena Blekinge project funded by Tillväxtverket and ERUF.

**Conflicts of Interest:** The authors declare no conflict of interest.

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
