# Peer review of "A Hybrid Data-Based and Model-Based Approach to Process Monitoring and Control in Sheet Metal Forming"

_processes, doi:10.3390/pr8010089_

Round 1

Reviewer 1 Report

The paper is interesting and generally, it deserves to be published with some revisions that are suggested below:

Article structure: Need to revise, e.g. Literature review need to build a  separate charter after Introduction.

Research purpose: Your study need to mention what the shortcomings of process monitoring and control approaches within sheet metal forming are and how can these be mitigated, e.g. on the Abstract. You may try to make a chart to show them.

Feedback mechanism: you may need a feedback mechanism on your framework to adjust the input factors in the process after used sensors  to measure it.

There are not so clear in your study to mention the four stages of Design Research Methodology (DRM) when the structured using the framework proposed.

Author Response

Please see the attachment. Note, that the document contains responses to both reviewers comments :-)

Reviewer 2 Report

The paper “A hybrid data- and model-based approach to process monitoring and control in sheet metal forming” describes the process monitoring aspects of sheet metal forming. This problem is especially relevant in the field of automotive industry. In my opinion, the article is suitable for the purposes of the Processes journal.

Title: The title of the paper is informative. It includes important terms and the message of the article.

Keywords: Keywords are well chosen.

Abstract: The abstract describes the context and provides a general picture of the methodological approach. The main outcomes are also described.

Introduction and literature review: Introduction defines the focus and explains the structure of the text. Literature review prepares the reader to understand the research part of the article. A summary table comparing the contributions could support the explanation.

Materials and Methods: In my opinion, the material and methods section is extensively discussed. One minor remark: what do you think about the digital twinning? Is it a potential solution for real-time monitoring and supervision in sheet metal forming?

Discussion, Conclusions and Future Research: Clear and adequate. Managerial implications should be added.

Author Response

Please see the attachment. Note,that the document contains responses to both reviewers comments :-)
